# Neoadjuvant Chemoradiotherapy in Locally Advanced Gastric Adenocarcinoma: Long-Term Results and Statistical Algorithm to Predict Individual Risk of Relapse

**DOI:** 10.3390/cancers17091530

**Published:** 2025-04-30

**Authors:** Miguel Ortego, Olast Arrizibita, Adriana Martinez-Lage, Ángel Vizcay Atienza, Laura Álvarez Gigli, Oskitz Ruiz, José Carlos Subtil, Maialen Zabalza, Victor Valentí, Ana Tortajada, María José Hidalgo, Onintza Sayar, Javier Rodriguez

**Affiliations:** 1Department of Medical Oncology, Clínica Universidad de Navarra, 31008 Pamplona, Spain; mortego@unav.es (M.O.); avizcay@unav.es (Á.V.A.); atortajadal@unav.es (A.T.); mhidalgo.7@unav.es (M.J.H.); 2Department of Mathematics and Statistic, NNBi, 31110 Noain, Spain; olast.arrizibita@nnbi.es (O.A.); oskitz.ruiz@nnbi.es (O.R.); maialen.zabalza@nnbi.es (M.Z.); onintza.sayar@nnbi.es (O.S.); 3Department of Radiation Oncology, Clínica Universidad de Navarra, 31008 Pamplona, Spain; amartinez-l@unav.es; 4Department of Pathology, Clínica Universidad de Navarra, 31008 Pamplona, Spain; lalvarezg@unav.es; 5Department of Gastroenterology, Clínica Universidad de Navarra, 31008 Pamplona, Spain; jcsubtil@unav.es; 6Department of GI Surgery, Clínica Universidad de Navarra, 31008 Pamplona, Spain; vvalenti@unav.es

**Keywords:** induction chemotherapy, preoperative chemoradiation, logistic regression, long-term outcome, locally advanced gastric cancer

## Abstract

Neoadjuvant chemoradiotherapy in locally advanced gastric cancer has been a matter of study in recent years, with the TOPGEAR trial as the referral for this scenery. We present the long-term oncological outcomes of a subset of patients treated with docetaxel-based induction chemotherapy, chemoradiation and surgery, and we aimed to identify subsets of patients who might benefit most from this approach.

## 1. Introduction

Gastric cancer (GC) is the fifth most diagnosed cancer and the third leading cause of cancer death worldwide [1]. The management of this disease is based on histopathological diagnosis complemented by molecular analysis; imaging techniques including endoscopic ultrasound, computed tomography (CT) and positron emission tomography (PET) in selected cases and diagnostic laparoscopy and peritoneal washings for cytology in locally advanced tumors. Once histological and staging data are obtained, evaluation from a multidisciplinary tumor board is critical. This process is essential for reaching a consensus on the best individualized therapeutic approach. It is recommended that the board includes medical oncologists, surgeons, gastroenterologists, radiologists, pathologists and, when possible, a nutritionist, among other involved professionals [2].

For the locally advanced stage, surgical resection with extended lymphadenectomy remains the only potentially curative treatment modality, at least in the early stages. Progression of the disease through the gastric wall and/or regional lymph nodes correlates with a high postoperative recurrence rate and a 5-year overall survival lower than 30% in Western countries [3,4]. This has led to the design of multimodal strategies for stages beyond IB disease, with most of them contributing to a variable gain in survival compared to surgery alone [5,6,7,8,9,10]. To date, there is a strong rationale to support using perioperative chemotherapy in locally advanced GC patients [11]. Although very recent data showed a lack of survival benefit when adding preoperative chemoradiation to perioperative chemotherapy [12], whether a subset of patients might benefit from this approach remains undetermined. Given preliminary promising data with the use of docetaxel-based triplet regimens in the neoadjuvant setting [13,14,15], we retrospectively assessed the long-term outcomes of a selected population of GC patients treated with docetaxel-based induction chemotherapy, chemoradiotherapy and surgery. Interpretable statistical algorithms that combine clinical, analytical and histopathological parameters were applied to rule out the individual risk of relapse following surgical resection.

## 2. Materials and Methods

### 2.1. Patient’s Characteristics

This retrospective analysis included patients with GC treated with induction chemotherapy (ICT) followed by preoperative chemoradiotherapy (CRT). The eligibility criteria included patients aged 18 to 80 years, with histologically confirmed gastric adenocarcinoma cT3-4 and/or cN+, and an Eastern Cooperative Oncology Group (ECOG) performance status of less than 2. Patients were excluded if they had a history of previous or coexisting cancer, non-adenocarcinoma histology, metastatic disease, or any clinically significant contraindication to receiving systemic chemotherapy. Furthermore, patients with tumors involving the gastroesophageal junction (GEJ) (Siewert I to III) were also excluded.

The pre-treatment evaluation involved documenting the patients’ medical history, conducting a full physical exam, performing routine laboratory tests, and obtaining a baseline CT scan and endoscopic ultrasound (EUS) with biopsy. A CT scan was also performed to evaluate the response at the end of ICT and prior to surgery. Patients were evaluated by a multidisciplinary board (which included medical and radiation oncologists, surgeons, gastroenterologists, radiologists, pathologists and nutritionists) with the most suitable therapeutic strategy being recommended on an individual basis, including the need for exploratory laparoscopy. Indications for surgery included the absence of metastatic disease, responding or stable disease after preoperative therapy and the possibility of achieving a macroscopically complete resection.

### 2.2. Procedures

All patients received preoperative docetaxel-based chemotherapy, either FLOT or DOX, on an outpatient basis with physical examination, blood tests, monitoring of the toxicity profile and regular clinical follow-up.

The radiation technique and target volume were tailored to each patient, considering the size and location of the primary tumor along with the regional lymph nodes affected. The clinical target volume encompassed the gross tumor volume, the whole stomach, and the draining locoregional lymph nodes (including perigastric, suprapancreatic, celiac, splenic hilar, porta hepatis, and pancreatoduodenal areas). The design and positioning of the radiation fields were meticulously planned to ensure that critical organs (such as the kidneys, heart, liver, and spinal cord) received doses within safe limits. Treatment planning was based on the recommendations of the International Commission on Radiation Units and Measurements (ICRUs). Generally, three fields using 15-MV photons were utilized to deliver 45 Gy over 5 weeks, with daily fractions of 1.8 Gy, five days per week. Seven coplanar, evenly spaced beams were used, with a variable number of segments in IMRT plans. Acute toxicity was evaluated according to the “Common Terminology Criteria for Adverse Events” version 5.0.

Throughout the course of CRT, all patients received simultaneous chemotherapy with capecitabine (650–825 mg/m² twice daily on the days of radiation). Weekly evaluations were conducted, including physical examinations, complete blood counts, renal function assessments, and monitoring and management of any therapy-related toxicities.

The radiological response was assessed using a CT scan at the end of ICT, followed by another CT scan and EUS 6–8 weeks after completing chemoradiation and before surgical resection.

A gastrointestinal pathologist performed the pathological examination of all surgical samples. A complete pathological response (pCR) was defined as the complete absence of any residual tumor in the surgical specimen (ypT0 ypN0). Histological regression of the primary tumor was evaluated according to the Becker criteria [16]. The nodal response was categorized using a four-point scale based on Smith et al. [17], as follows: Grade A—negative lymph node (LN) with no signs of preoperative therapy effect; Grade B—infiltrated LN with no observable effect from preoperative therapy; Grade C—infiltrated LN showing partial histological regression due to preoperative therapy; Grade D—complete pathological response in an infiltrated LN. A favorable pathological response was defined by a combination of Becker grade 1a or 1b responses along with grade D nodal regression. Downstaging was determined by a reduction in pathological T and/or N stage (ypTNM) when compared to the initial clinical staging (cTNM).

The follow-up plan involved regular physical examinations and comprehensive laboratory tests, along with CT scans, scheduled every four months during the first two years, then every six months for the subsequent two years, and once a year thereafter.

### 2.3. Clinical Information

The following factors were retrospectively evaluated for each patient: sex, age, ECOG, primary tumor site, clinical stage (pTNM), Lauren classification, type of chemotherapy, number of cycles received, dosage, dose intensity (mg/m^2^/week), toxicity profile, radiological response, presence of vascular and perineural invasion, endogenous pyrimidine levels, area under the curve of 5-fluorouracil, type of surgical intervention, pathological stage (pTNM), number of resected lymph nodes, tumor regression grade, surgical margins, date and type of disease relapse (local and/or distant) and use of subsequent treatments. Additionally, liver and kidney function, complete blood count, coagulation profile, as well as the neutrophil-to-lymphocyte ratio (NLR) and neutrophil-to-platelet ratio (NPR) were assessed from the start of neoadjuvant treatment until surgery.

### 2.4. Statistical Analysis

Statistical analyses were initially performed using R software (Rstudio, R version 3.6.1 2019-07-05) and SPSS v.20. For quantitative variables measured over time, new variables were created by calculating the maximum, minimum, standard deviation, mean, median, and the difference between the initial (baseline) and final (pre-surgery) values (referred to as the differential estimator, or esti-differ). The analysis was carried out in three stages: (1) clean-up and general analysis (descriptive analysis, detection of outliers and missing values and testing of basic assumptions); (2) correspondence analysis to reduce the number of categories in some of the categorical variables and (3) principal component analysis (PCA). These steps provided a comprehensive overview of the data, its structure and internal relationships. The Mahalanobis distance method was used to identify potential outliers in the dataset [18,19]. Specifically, individuals with outliers were identified as those which Mahalanobis distance exceeded the limit (*p* > 0.001) set by the distance distribution.

A locoregional recurrence was defined as the presence of disease in the anastomosis site, gastric remnant, or abdominal lymph nodes, including regions such as the porta hepatis, celiac axis, superior mesenteric artery/vein, para-aortic, and gastrohepatic nodal areas. Progression-free survival (PFS) was calculated from the beginning of ICT until disease relapse (local and/or distant), death from any cause, or the last follow-up without recurrence. Overall survival (OS) was measured as the time from the start of neoadjuvant treatment until either death from any cause or the last recorded contact with a living patient. Both OS and PFS were estimated using the Kaplan–Meier method.

This retrospective observational study was approved by the Clinical Research and Ethics Committee of Navarra (registration number 2021.060).

### 2.5. Variable Creation

For this research, the logistic regression (LR) method was selected to design the predictive model, as the objective was to achieve an interpretable model that enable us to certainly understand their performance, allowing us to verify each of their predictions and to comprehend the rational for these forecasts. Compared to “black box” models, interpretable parameters are generally preferred in medical practice [20].

After completing the statistical analysis, we obtained a valid dataset that allowed us to begin the process of constructing the predictive model by identifying the variables most statistically significant in predicting relapse, using logistic regression (LR) with a univariate test. Next, we created a series of models by systematically adding and removing predictor variables until we found the best model fit with the least complexity.

To select the optimal model, we performed ANOVA tests on models containing different numbers of variables, including the variables with the highest univariate statistical significance, ranked by their importance. This process enabled us to compare the predictive performance of each model and determine whether adding new variables would significantly improve the model’s predictive accuracy. The overall fit of each model was assessed using the likelihood ratio chi-square statistic to determine if it was significantly better than the previous version.

The predictive model was validated using the stratified k-fold cross-validation method. The dataset was divided into k subsets, ensuring that the division was random but that the proportion of patients in each class (those who relapsed and those who did not) remained consistent across the subsets. For each iteration, k − 1 subsets were used for training, while the remaining subset was used for testing the model. This process was repeated k times, covering all possible combinations of the subsets, and the arithmetic mean of the results was used as the final estimate of model performance. After cross-validation with the selected final model, an additional validation step was performed using an independent dataset of 19 patients with locally advanced gastric cancer. These patients met the same inclusion criteria as those in the original sample, ensuring consistency in the validation process.

## 3. Results

From 2010 to 2019, 48 GC patients were included in the analysis, with most patients being men (79%). The median age at the time of diagnosis was 61 years (range, 36–76). The EUS stage was cT4 in 11 patients (23%) and cN+ in 41 patients (85%). The baseline patient and tumor characteristics are shown in Table 1.

### 3.1. Preoperative Therapy

During ICT, 6 (12.5%) and 42 (87.5%) patients received FLOT and DOX, respectively. The median number of chemotherapy cycles administered was 3 (range 2–6). Treatment-related toxicities are summarized in Table A1 (Appendix A). Treatment delays and dose reductions were required in 8.3% and 10.4% of the patients, respectively. Grade 3–4 toxicity included anemia in four patients (8.4%), who required blood products transfusion, neutropenia (12.6%) and fatigue (8.4%). Three patients required hospital admission, one due to pneumonia and the other two due to grade 4 febrile neutropenia. Seven patients received G-CSF support. No other grade 4 toxicities were observed, and no chemotherapy-related deaths were reported.

Forty-six patients (96%) completed the initially planned radiotherapy schedule. Radiation was stopped in two patients due to fatigue and anorexia. The mean dose of radiotherapy administered was 44.9 Gy with a median treatment length of 34 days (range, 22–43). Most frequent grade 3 toxicity included fatigue (10.4%) and anorexia (8.3%). Four patients (8%) required admission to hospital: three due to grade 3 fatigue and anorexia, and one patient due to a cardiac insufficiency (2%). Total parenteral nutrition was administered in three patients. No grade 4 toxicities were seen.

### 3.2. Surgery and Pathological Evaluation

Forty-six patients (96%) underwent surgery. In two patients, unresectability criteria were found in the operating room despite radiological stable disease. In thirty-four patients, gastrectomy was performed via laparotomy, and in twelve of them, it was performed via laparoscopy. Subtotal and total gastrectomy were performed in 17 (35.4%) and 29 (60.4%) patients, respectively. Lymph node dissection was D2 in 43 and D1+ in 3 patients. The median number of harvested nodes was 12 (range 0–30), in line with other works that suggested a reduction in the number of retrieved lymph nodes with the use of preoperative radiotherapy in gastric cancer patients [21]. An R0 resection was achieved in 43 patients (89.6% of the patients on an intent-to-treat basis). Three patients presented serious postoperative complications and required re-intervention due to a small bowel perforation (one patient), bleeding from the mesocolon surface (one patient) and suture dehiscence (one patient).

The histopathological characteristics of the surgical specimens are described in Table A2 (Appendix A). A complete pathological response (ypT0ypN0) was achieved in nine patients (18.8%). Thirty patients (62.5%) had tumor-free LNs, including eighteen patients (37.5%) with a complete nodal pathological response (grade D). A favorable pathological response (Becker 1a,b and nodal grade D) was achieved by 19 patients (39.6%). Downstaging was observed in 34 patients (70.8%).

### 3.3. Survival Analysis

After a median follow-up of 49.5 months (range, 4.6–212.3), the median PFS and OS were 26.8 (95% CI, 17.2–80.1) and 51.3 months (95% CI, 26.4–131.6), respectively. The five-year actuarial PFS and OS were 44% and 48%, respectively. Thirty-one patients have died, 10 without disease recurrence (three patients due to second neoplasms, one due to stroke, two from postoperative complications, one from non-malignant intestinal obstruction and three other patients years later due to chronic conditions). At the time of analysis, 17 patients (35%) were alive and disease-free.

Twenty-one patients (44%) relapsed, with 52%, 29% and 10% of recurrences occurring during years 1, 2 and 3 after surgery, respectively. Patterns of relapse included a local-only relapse in 6 patients, systemic relapse in 12 patients and both local and systemic relapse in 3 patients. Disease relapse was more frequent in lymph node-positive patients (ypN0 20%; ypN+ 81%; *p* < 0.001). Sites of distant relapse included the liver (three patients), lymph nodes (two patients), lung (one patient) and peritoneal carcinomatosis (nine patients).

Patients who achieved downstaging had 5-year PFS (47% vs. 42%; *p* = 0.51) and OS (53% vs. 42%; *p* = 0.74) similar to those without downstaging. PFS and OS were numerically longer but not statistically different for the subset of patients who achieved a pCR. The 5-year actuarial PFS and OS rates for Becker 1a,b were like Becker 2,3 (5-year PFS: 50% vs. 40%; *p* = 0.56; 5-year OS: 58% vs. 40%; *p* = 0.37).

Five-year actuarial PFS was significantly longer in ypN0 patients compared to ypN+ patients (60% vs. 18%, *p* = 0.012), as was the 5-year actuarial OS (63% vs. 25%; *p* = 0.029). Patients with truly negative nodes from the baseline (grade A) and those with a nodal complete response (Grade D) had similar 5-year survival rates (PFS 58% vs. 61%; *p* = 1; OS 67% vs. 61%; *p* = 1). Patients with Becker 1a,b and nodal regression grades D and A experienced a survival advantage in terms of 5-year PFS (63% vs. 31%; *p* = 0.039) and OS (68% vs. 34%; *p* = 0.038) when compared to the rest of the patients.

### 3.4. Population Model Development

After the process of cumulative variable selection, those included in the model are highlighted in Table 2. In addition, Table 3 displays the statistical significance of each individual variable included in the model. The strength of adjustment was assessed using Mcnemar’s pseudo R2. A better model is indicated by higher values, which range from 0 to 1. The optimal logistic regression was chosen using a combination of this statistic, the numerical significance of the variables and ANOVA comparisons between models of varied sizes. A Mcnemar’s pseudo R2 of 0.78 was obtained for the final model. Six-times cross-validation was used after the model had been created. The accuracy, sensitivity and specificity scores (mean ± sd) for cross-validation of the model were 0.79 ± 0.12, 0.74 ± 0.21 and 0.88 ± 0.14, respectively (Figure 1).

### 3.5. Model Interpretation

Through the value obtained using the odds ratio, it is feasible to evaluate the impact of each variable included in the model on the predictions generated by the algorithm.

For LN grade C, while keeping all other variables constant, the relative probability the outcome (relapse) increased by approximately 94,300% compared to the reference LN grade A. However, the *p*-value (0.0977) is marginally significant, suggesting that this result should be interpreted with caution.For LN grade B, while keeping all other variables constant, the relative probability of a relapse increased by approximately 125,500% compared to LN grade A. This effect is statistically significant, as indicated by a *p*-value of 0.0227.For LN regression grade D, the effect on the risk of relapse is comparable to that of LN grade A. The *p*-value is not significant, suggesting insufficient evidence to support a difference between these categories.For the histological intestinal type, while keeping all other variables constant, the relative probability of relapse decreased by approximately 97.8% compared to the diffuse category. This result is statistically significant, with a *p*-value of 0.0482.A one-unit increase in alkaline phosphatase differential estimator would reduce the odds of relapse by 7.7%. This result is statistically significant, with a *p*-value of 0.0473.A one-unit increase in a hematocrit media estimator would reduce the odds of relapse by 44.6%. This result is statistically significant, with a *p*-value of 0.0322.

### 3.6. Validation of the Model

A sample of 19 potentially resectable LAGC patients from our institution with similar clinical characteristics was used for model validation. Eight of the nineteen patients in this cohort relapsed. Table 4 and Table 5 summarize the descriptive analysis of the variables in the validation model dataset. In 15 out of 19 patients, the model properly predicted the real outcome. The model’s predictive accuracy at the individual level was 79%, with a sensitivity and a specificity of 88% and 73%, respectively.

## 4. Discussion

Arguments favoring preoperative RT are increased patient compliance, resectability and R0 rates, smaller radiation target volumes and reduced displacement of contiguous structures, leading to decreased treatment toxicity [22,23,24,25]. The main drawbacks of this strategy are the possibility of tumor progression during neoadjuvant treatment and the delay of curative surgery in case of severe toxicity. Recent results from the TOPGEAR trial [11] comparing preoperative chemoradiotherapy plus perioperative chemotherapy or perioperative chemotherapy alone showed no significant differences in terms of survival times between arms. Nonetheless, some post hoc analyses are awaited (Lauren histological subtypes, use and type of systemic therapy upon relapse, impact of unbalances in the number of chemotherapy cycles received between arms, molecular subgroups, etc.). In the present work, we aimed to report the long-term oncological outcomes of locally advanced gastric adenocarcinoma patients treated with docetaxel-based induction chemotherapy, chemoradiation and surgery, when feasible, and to identify subsets of patients who might benefit most from this approach.

A R0 rate ranging 90%, a pathological complete response of 18% and a complete nodal response (grade D) of 37% translated into 5-year progression-free and overall survival rates of 44% and 48%, respectively. Moreover, a median OS of more than 4 years was achieved, even though almost 90% of the patients were EUS-N+, 15% had T4 tumors and 60% had a poorly differentiated histology.

Although the study has the proper limitations of its retrospective nature, the small sample size, the potential biases in treatment selection and the fact that it was conducted at a single institution, the long follow-up period confirmed, in line with other authors [26], that, with this multimodal approach, almost 50% of the patients did not relapse 5 years after surgery. This follow-up period also allows describing the patterns of disease relapse in the long term. Non-randomized studies assessing preoperative radiotherapy have demonstrated an improvement in the local relapse rate, but locoregional recurrences are still reported in 13–42% of the patients [27]. Consistent with these findings, 20% of our patients had a locoregional relapse. Interestingly, ypN+ patients had a higher likelihood of such a relapse, reinforcing the correlation between a good pathological response and a lower locoregional recurrence [28,29,30]. Most local relapses occurred within the first two years after surgery and were usually accompanied by a systemic relapse. No local relapses were observed after 3 years from surgery, and no systemic relapses were observed 4 years after surgery. If these results are confirmed in larger and prospective trials, they may be of help in the design of risk-adapted surveillance programs.

One of the major issues of concern with the use of preoperative radiation therapy remains toxicity. Trials evaluating preoperative radiation using either IMRT or 3D plans have reported grade 3–4 adverse events in approximately 50% of the patients, along with a hospitalization rate of 28%. Although no grade 4 toxicity was observed, almost 30% of the patients required admission to a hospital, most of them during the interval between the end of radiotherapy and surgery. On the other hand, rates of morbidity and mortality in the present analysis were 5% and 3%, respectively. We have carefully reviewed the dose volume histograms of the small bowel and mesocolon in patients with severe postoperative complications. The estimated mean doses TD5/5 for small bowel irradiation in each patient were 18.54 Gy (patient with mesocolon bleeding), 40.5 Gy (patient with small bowel perforation) and 49.4 Gy (patient with suture dehiscence). The mean doses administered were consistent with the ICRU (International Commission on Radiation Units and Measurements) recommended constraints. We thus feel that these complications were unexpected and may not be related to the radiation field or dose. Indeed, morbidity and mortality rates of 7–47% and 3–10%, respectively, have been described with preoperative strategies [31]. Nevertheless, these data point to the need for accurate patient selection, as well as close clinical monitoring of side effects by a multimodal and experienced team.

Artificial intelligence may offer an alternative method for improving our predictive capacity for cancer-related events [32]. Although alkaline phosphatase and hematocrit levels are known prognostic factors in patients with gastric cancer [33,34,35,36], the two most relevant variables according to our model are the Lauren histological subtype and the degree of LN regression. Each of these variables significantly influenced the model predictions, either increasing or decreasing the probability of a relapse. The Lauren subtype is a well-recognized predictor of poor outcome and therapy resistance [37,38]. In a phase III trial of 182 LAGC patients comparing preoperative chemoradiotherapy vs. chemotherapy [39], those patients with intestinal-type tumors included in the chemoradiotherapy arm had a higher likelihood of achieving a complete nodal regression than those with a diffuse or mixed subtype, with no such difference being found in the chemotherapy-only arm. In the ARTIST trial, adjuvant chemoradiotherapy did not significantly improve the OS compared to chemotherapy alone after a D2 dissection. However, subgroup analysis showed that chemoradiotherapy significantly improved DFS in patients with node-positive disease and intestinal type histology (3-year DFS for intestinal type of 94% vs. 83% in favor of chemoradiation) [40]. In the ARTIST-2 [10], the hazard ratio for recurrence for the intestinal and diffuse subtypes was 0.45 and 0.81, respectively. The long-term analysis of the INT-0016 also suggested that chemoradiotherapy did not appear to confer a benefit in patients with diffuse histology [41]. In our model, compared with the Lauren diffuse subtype, the presence of an intestinal subtype was also associated with a significant reduction in the risk of relapse.

Lymph node involvement and nodal regression are well-known prognostic factors for recurrence and long-term survival in resected GC patients [42]. In accordance with our previous findings [43,44,45,46], we reinforced that the residual pathological nodal status, rather than graded histological response of the primary tumor, is a prognostic factor for a longer OS. Compared to grade A, both grades C and B were associated with a significant increase in the risk of relapse. Interestingly, LN regression grade D showed the same effect on the risk of relapse as grade A. In fact, we found no differences in the 5-year PFS and OS rates between patients with these two grades of nodal regression. The 5-year OS in this subgroup is remarkable and makes ypN0, either because the patient has a truly negative LN from baseline or due to the achievement of a complete nodal pathological response after preoperative therapy, a good surrogate marker for a long-term outcome.

## 5. Conclusions

In conclusion, for LAGC patients, induction chemotherapy followed by preoperative chemoradiation offers a good chance at achieving favorable histopathological features that may translate into a prolonged long-term outcome. Our four-variable model was able to predict the individual risk of relapse with accuracy, sensitivity and specificity scores of 0.82, 0.59 and 0.96, respectively. However, the small sample size restricts generalizability, and both careful interpretation and an external validation are required before firm conclusions can be drawn. Considering previous findings of a higher likelihood of nodal regression with the use of preoperative chemoradiotherapy [43,47,48] and the remarkable 5-year OS in the ypN0 subgroup, further research through multicenter trials with this approach in fit patients with intestinal-type histology seems warranted.

## Figures and Tables

**Figure 1 cancers-17-01530-f001:**
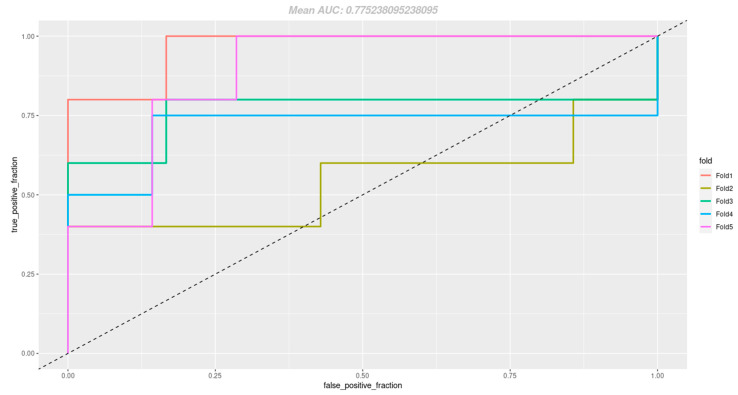
Receiver operating characteristic (ROC) curves of the logistic regression (LR) model’s predictive ability for the 6-fold cross-validation. The LR model was trained to predict the target variable, which is a dichotomous variable representing the risk of relapse or no risk of relapse. To achieve this goal, the model employs the predictor variables listed in Table 2.

**Table 1 cancers-17-01530-t001:** Patient characteristics (N = 48).

Characteristic	Patients (%)
Age (median, range)	61 (36–76)
Gender	
Male	38 (79%)
Female	10 (21%)
ECOG performance status	
0	7 (15%)
1	38 (85%)
Location	
Antrum	18 (37.5%)
Body	25 (52.1%)
Pylorus	1 (2.1%)
Fundus	4 (8.3%)
EUS-T stage	
cT3	37 (77%)
cT4a	8 (16.7%)
cT4b	3 (6.3%)
EUS-N stage	
cN0	7 (15%)
cN+	41 (85%)
Histologic grade	
Well differentiated	1 (2%)
Moderately differentiated	22 (46%)
Poorly differentiated	25 (52%)
Linitis plástica	
Yes	9 (18.8%)
No	27 (81.3%)
Lauren Histologic classification	
Diffuse	22 (46%)
Intestinal	26 (54%)
Baseline EUS	48 (100%)
Baseline CT-scan	48 (100%)
Exploratory Laparoscopy	
Yes	23 (47.9%)
No	25 (52.1%)
ICT regimens *	
FLOT	6 (12.5%)
DOX	42 (87.5%)
Radiotherapy technique	
3D	37 (77.1%)
IMRT	10 (20.8%)
Unknown	1 (2.1%)
Radiotherapy dose (mean, range)	44.9 (38–50) Gy

* FLOT consisted of docetaxel 50 mg/m^2^, oxaliplatin 85 mg/m^2^, leucovorin 200 mg/m^2^ and fluorouracil 2600 mg/m^2^. DOX consisted of docetaxel 60 mg/m^2^, oxaliplatin 85 mg/m^2^ and capecitabine bid 650–825 mg/m^2^ every 3 weeks.

**Table 2 cancers-17-01530-t002:** Features included in the model.

Variables	Statistic	Value
Alkaline phosphatase differential estimator	Min	−48
	Mean	19.91
	Median	8.5
	Max	213
Mean hematocrit	Min	29.55
	Mean	37.51
	Median	37.79
	Max	48.6
LN regression grade	A	9 (23.7%)
	C	4 (10.5%)
	B	10 (26.3%)
	D	15 (39.5%)
Lauren subtype	Diffuse	20 (52.6%)
	Intestinal	18 (47.4%)

**Table 3 cancers-17-01530-t003:** Statistical significance of the variables included in the model.

Variables	Estimate	*p*-Value	ODDS Ratio
LN regression C (ref: A)	6.85083	0.0977	944
LN regression B (ref: A)	7.13547	0.0227 *	1255.727
LN regression D (ref: A)	1.98599	0.2985	7.286257
Lauren_Intestinal (ref: Diffuse)	−3.82001	0.0482 *	0.02192758
Alkaline phosphatase differential estimator	−0.08061	0.0473 *	0.9225534
Mean hematocrit	−0.58974	0.0322 *	0.5544714

* statistically significant (*p* < 0.05).

**Table 4 cancers-17-01530-t004:** Basic characteristics of the validation dataset that were included in the predictive model.

Variables	Statistic	Value
Alkaline phosphatase differential estimator	Min	0
	Mean	20.12
	Median	9
	Max	147
Mean hematocrit	Min	30.98
	Mean	36.68
	Median	36.61
	Max	42.06
LN regression grade	A	5 (% 26.3)
	B	2 (% 10.5)
	C	7 (% 39.9)
	D	5 (% 26.3)
Lauren	Diffuse	7 (% 36.8)
	Intestinal	12 (% 63.2)

**Table 5 cancers-17-01530-t005:** Confusion Matrix.

**Real**		**Predicted**	
		**No**	**Yes**
	**No**	8	3
	**Yes**	1	7

## Data Availability

The original contributions presented in this study are included in the article. Further inquiries can be directed to the corresponding author.

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
