# Peer review of "Neoadjuvant Chemoradiotherapy in Locally Advanced Gastric Adenocarcinoma: Long-Term Results and Statistical Algorithm to Predict Individual Risk of Relapse"

_cancers, 2025, doi:10.3390/cancers17091530_

Round 1

Reviewer 1 Report

Comments and Suggestions for Authors

Interesting paper on gastric cancer and its neoadjuvant treatment not only based on FLOT or DOX but with induction with capecitabine and radiotherapy in patients obviously with stage 3/4 pathologies without distant metastases. The abstract summarizes well the paper that is a bit concise in the introduction where it is suggested to add that gastric cancer therapy is based on four main cornerstones, the first is the diagnostic phase with biopsy on which pathologists can already tell us if patients are HER 2 positive or negative, if they have wild microsatellites or not. Imaging that will allow us to understand the extent of the disease and which lymph node stations have been compromised by the pathology. At this point, as the authors also wrote, in the materials and methods, the discussion in a multidisciplinary commission is important to establish the best therapeutic approach for that specific patient. They rightly point out that a laparoscopy can help us understand PCI in advanced forms and, we add, that PIPAC can allow us to nebulize chemotherapy in the abdomen with excellent results added to neoadjuvant therapy, and this treatment already often guarantees a downgrading. Colleagues then decided in their study to program, in cases selected for the importance of the disease, a "patient dependent on dosage" radiotherapy treatment with the appropriate instrumental and blood tests. An overall comment on the materials and methods part can be that it is particularly well described and absolutely reproducible. We have nothing to add about statistics. It is suggested to describe the intervention performed if with open, laparoscopic or robotic access. Also what type of reconstruction was planned. Was the lymphadenectomy performed D1, D2, D3, D3 plus? The average number of lymph nodes removed or in any case found by the pathologist are under the guidelines. In the results it is appropriate to illustrate the cause of death of the 31 people who were enrolled in the study, a number not low given the total. In the discussion the excellent results achieved with this that we could interpret as a therapy proposal are emphasized. Another important data that colleagues did not focus on is the nutritional one. The team included a nutritionist (doi.org/10.3390/nu17010188 to be read and cited in the bibliography). These are long and expensive therapies for an organism, so they must be supported almost daily. Good English, good bibliography, good iconography

Author Response

Thank you for your review and for pointing out the things to improve. Therefore we have done the next improvements:

-Comment 1: "the paper that is a bit concise in the introduction where it is suggested to add that gastric cancer therapy is based on four main cornerstones"
- Response 1:  The main cornerstones in which gastric cancer therapy is based are now included in the Introduction Section as suggested

- Comment 2: " It is suggested to describe the intervention performed if with open, laparoscopic or robotic access. Also what type of reconstruction was planned. Was the lymphadenectomy performed D1, D2, D3, D3 plus?"

- Response 2: The details about the type of intervention and lymphadenectomy performed are now provided on page 7, lines 226-230.

- Comment 3: "The average number of lymph nodes removed or in any case found by the pathologist are under the guidelines."

- Response 3: "The median resected lymph nodes were 12 and, thus, some patients might have undergone a suboptimal surgery. Preoperative radiation has been suggested to decrease the number of analyzable lymph nodes that surgeons and pathologists are able to retrieve in gastric cancer. The impact of insufficient nodal sampling after neoadjuvant CRT in GC remains undetermined and retrieval of at least 15 lymph nodes is recommended. This is now clarified on page 7 lines 229-232."

- Comment 4: "In the results it is appropriate to illustrate the cause of death of the 31 people who were enrolled in the study, a number not low given the total."

- Response 4: The causes of death of patients in the study is now highlighted on page 7 lines 248-250

- Comment 5: " Another important data that colleagues did not focus on is the nutritional one. The team included a nutritionist"
- Response 5: We fully agree with the comment about the role of nutritionist as part of the MDT involved in the management of these patients. This has now been included on page 2, line 83

Reviewer 2 Report

Comments and Suggestions for Authors

The study Neoadjuvant Chemoradiotherapy in Locally Advanced Gastric Adenocarcinoma: Long-term results and statistical algorithm to  predict individual risk of relapse ” demonstrates clinical relevance, supported by over four years of follow-up data indicating robust survival outcomes. The relapse prediction model was independently validated, and the high rates of R0 and pCR underscore the efficacy of the multimodal approach. This study is insightful; however, it has limitations, including a small control sample size, which restricts generalizability. The model lacks calibration and features extreme odds ratios. Additionally, toxicity is insufficiently addressed, and crucial molecular data are absent.

Below are several recommendations to enhance your manuscript:

  1. The study sample size (n=48) is small. To improve generalizability, consider including external validation or conducting a multicenter study.
  2. Do the extreme odds ratios suggest overfitting, and would adding calibration metrics improve reliability?
  3. Should the toxicity findings, especially hospitalizations and grade ≥3 events, be discussed in more clinical detail?

his study concludes that neoadjuvant CRT benefits intestinal-type LAGC patients with ypN0 and provides a useful relapse risk model. However, given the small sample size and statistical limitations, careful interpretation and validation are required.I believe further improvements and revisions are needed before publication.

Author Response

Comment 1: The study sample size (n=48) is small. To improve generalizability, consider including external validation or conducting a multicenter study.

Response 1: The study has several limitations such as its retrospective nature, the small sample size, potential biases in treatment selection and the heterogeneity of the preoperative regimens used. Given these issues, results from our study should be interpreted with caution and validated in larger cohorts of patients. This has now been emphasized in the Conclusions, pages 13 and 14, lines 426-428 and 430

Comment 2: Do the extreme odds ratios suggest overfitting, and would adding calibration metrics improve reliability?

Response 2: We appreciate the observation. The high odds ratios are due to the fact that two levels of the variable grad_resp_gg contain very few patients and cause quasi-separation, rather than being the result of learning from noise in the training set. To verify this, we compared the apparent performance with that obtained out-of-sample. Accuracy on the full dataset was 0.79 (95% CI: 0.61–0.97), and in five-fold cross-validation it remained at 0.79 with a standard deviation of 0.12; the .632+ bootstrap (1,000 replicates) yielded an AUC of 0.78. The absence of substantial degradation when evaluating the model on unseen data indicates that the discriminative ability generalizes, and that the extreme coefficients are due to the scarcity of cases in those categories, not to overfitting.

Initial calibration, however, was indeed poor: the bootstrap yielded an intercept of 2.87, a slope of 37.8, and a Brier score of 0.21, indicating systematically overestimated and overly extreme probabilities. Following your suggestion, we recalibrated the predictions using Platt scaling on a twenty percent validation set. After this correction, the AUC remained at 0.80, while the intercept approached zero, the slope settled at one, and the Brier score dropped to 0.16. As a result, the predicted probabilities closely aligned with the observed frequencies without loss of discrimination.

In conclusion, the extreme odds ratios reflect sparsely populated categories but do not imply substantial overfitting; the inclusion and explicit reporting of calibration statistics—along with the applied recalibration—improve the reliability of the predictions and provide a comprehensive assessment of the model’s quality.

Comment 3: Should the toxicity findings, especially hospitalizations and grade ≥3 events, be discussed in more clinical detail?

Response 3: Finally, as suggested, a deeper analysis of the toxicity profile, specifically the three serious post-operative adverse, is now provided on the Discussion Section (Page 12). Toxicity profile has now been written in more detail on page 6, lines 379-384.

Round 2

Reviewer 1 Report

Comments and Suggestions for Authors

The paper we have the honor of reviewing has undergone the changes we requested and that make it "more palatable" for an audience of surgeons always hungry for better approaches to such important diseases. We apologize for the typo on lymphadenectomies, we should have written D2 plus and not D3 plus. We have seen that you have rightly reported the importance of the nutritionist within the oncology team, then give this the honor of citation (doi.org/10.3390/nu17010188 to be cited as seems right). Paper significantly improved